# Genetic and Epigenetic Regulation in *Lingo-1*: Effects on Cognitive Function and White Matter Microstructure in a Case-Control Study for Schizophrenia

**DOI:** 10.3390/ijms242115624

**Published:** 2023-10-26

**Authors:** Jessica L. Andrews, Andrew Zalesky, Shalima Nair, Ryan P. Sullivan, Melissa J. Green, Christos Pantelis, Kelly A. Newell, Francesca Fernandez

**Affiliations:** 1School of Medical, Indigenous and Health Sciences, and Molecular Horizons, Faculty of Science, Medicine and Health, University of Wollongong, Wollongong, NSW 2522, Australiaknewell@uow.edu.au (K.A.N.); 2Melbourne Neuropsychiatry Centre, Department of Psychiatry, The University of Melbourne, Carlton South, VIC 3053, Australia; azalesky@unimelb.edu.au (A.Z.); cpant@unimelb.edu.au (C.P.); 3Epigenetics Research Program, Genomics and Epigenetics Division, Garvan Institute of Medical Research, Sydney, NSW 2010, Australia; s.nair@garvan.org.au; 4ARC Centre of Excellence for Electromaterials Science, Intelligent Polymer Research Institute, AIIM Facility, University of Wollongong, Wollongong, NSW 2522, Australia; ryan.sullivan@sydney.edu.au; 5School of Clinical Medicine, Discipline of Psychiatry and Mental Health, UNSW Sydney, Sydney, NSW 2052, Australia; melissa.green@unsw.edu.au; 6School of Behavioural and Health Sciences, Faculty of Heath Sciences, Australian Catholic University, Banyo, QLD 4014, Australia; 7Healthy Brain and Mind Research Centre, Australian Catholic University, Fitzroy, VIC 3065, Australia

**Keywords:** Lingo-1 gene, schizophrenia, single nucleotide polymorphism, methylation, white mater integrity, cognition

## Abstract

Leucine-rich repeat and immunoglobulin domain-containing protein (Lingo-1) plays a vital role in a large number of neuronal processes underlying learning and memory, which are known to be disrupted in schizophrenia. However, Lingo-1 has never been examined in the context of schizophrenia. The genetic association of a single-nucleotide polymorphism (SNP, rs3144) and methylation (CpG sites) in the *Lingo-1* 3′-UTR region was examined, with the testing of cognitive dysfunction and white matter (WM) integrity in a schizophrenia case-control cohort (n = 268/group). A large subset of subjects (97 control and 161 schizophrenia subjects) underwent structural magnetic resonance imaging (MRI) brain scans to assess WM integrity. Frequency of the rs3144 minor allele was overrepresented in the schizophrenia population (*p* = 0.03), with an odds ratio of 1.39 (95% CI 1.016–1.901). CpG sites surrounding rs3144 were hypermethylated in the control population (*p* = 0.032) compared to the schizophrenia group. rs3144 genotype was predictive of membership to a subclass of schizophrenia subjects with generalized cognitive deficits (*p* < 0.05), in addition to having associations with WM integrity (*p* = 0.018). This is the first study reporting a potential implication of genetic and epigenetic risk factors in *Lingo-1* in schizophrenia. Both of these genetic and epigenetic alterations may also have associations with cognitive dysfunction and WM integrity in the context of the schizophrenia pathophysiology.

## 1. Introduction

During the last decade, the Lingo-1 signalling pathways have been extensively reported to modulate neuron growth and survival, oligodendrocyte differentiation, and myelination [1]. Consequently, Lingo-1 dysfunction has been implicated in a number of neurological and psychiatric disorders, such as Alzheimer’s disease and multiple sclerosis [1]. Its inhibitory role in myelination and neuronal growth and survival, has led to the development of Lingo-1 antagonists (such as engineered antibody), tested in vitro and in vivo as a potential therapy for demyelinated diseases such as multiple sclerosis [1]. Disruption to *Lingo-1* OMIM (609791) has previously been associated with Parkinson’s disease and essential tremor [2,3,4,5] but not with multiple sclerosis [6]. However, no further explorations have been undertaken to characterize *Lingo-1* associations with other disorders. We previously provided the first study reporting higher protein levels of Lingo-1 in the dorsolateral prefrontal cortex (DLPFC), a critical region for learning and memory processes, in schizophrenia sufferers compared to controls [7]. Considering this prior finding and the etiological background of schizophrenia, resulting from a combination of both genetic and environmental factors [8,9], the present study examined the potential association between genetic and epigenetic alterations in *Lingo-1* with schizophrenia, and their implications for cognitive impairment and WM integrity

Here, we examined the association of rs3144 and surrounding CpG sites within the 3′-UTR region of *Lingo-1*, known to mediate gene expression at both transcriptional and post-transcriptional levels [10] and consequently affecting protein expression. Furthermore, 3′-UTR is the primary site of action for regulatory microRNA and proteins, and it has a high density of methylation sites with potential functional roles [11,12,13]. In a genome-wide study of deoxyribonucleic acid (DNA) methylation patterns across 17 human somatic tissues, the majority of hypermethylated CpGs were located in the 3′-UTR [13]. Additionally, a high density of CpGs sites were found at the boundaries of gene coding regions, including the 3′-UTR, which are fundamental for appropriate gene translation [14]. Due to Lingo-1’s role in regulating neuronal processes underlying cognition, and considering that cognitive deficits are present in 75–85% of schizophrenia patients [15], we performed analyses to uncover potential associations between *Lingo-1* genetic and epigenetic markers using cognitive function tests. Since cognitive dysfunction has been associated with decreased fractional anisotropy (FA) [16,17,18], an index of WM integrity reflecting axonal fibre density, diameter, and myelination [19], FA was also examined for associations with these tested genetic and epigenetic markers. For the first time, this study reports novel genetic and epigenetic risk factors in *Lingo-1*, as related to cognitive dysfunction and WM integrity in schizophrenia.

## 2. Results

### 2.1. Rs3144 and Methylation Association with Schizophrenia

Rs3144 did not deviate from HWE (*p* > 0.05). There was a significant association between the allelic frequency of rs3144 and schizophrenia (*p* = 0.03; Table 1). Sex specific analysis revealed that this significant association remained in the male schizophrenia subjects (*p* = 0.04). There was no significant allele carrier or genotypic association between this SNP and schizophrenia when all subjects were considered, nor when males and females were analysed separately.

Two-way MANOVA for diagnosis and genotype showed that, when all subjects were considered (i.e., both males and females), there was a significant effect of genotype on methylation status at CpG8 (*p* = 0.041; see Appendix A), with post hoc analyses revealing that the percentage of methylation at this CpG site was higher in TT and TC genotype subjects compared to CC individuals (*p* = 0.033 and *p* = 0.012, respectively). Furthermore, two-way MANOVA for diagnosis and allele carriers (TT:TC/CC) revealed a significant diagnosis effect on methylation status at CpG5 in all subjects (*p* = 0.032). Controls had a higher percentage of methylation at this CpG site than schizophrenia subjects (*p* = 0.032). A similar effect was observed in female subjects at CpG10 (*p* = 0.013). Controls were found to have a higher methylation percentage compared to schizophrenia subjects (*p* = 0.013).

### 2.2. Rs3144 and Methylation Effects on Cognition

Two-way MANOVA (diagnosis (or allele carrier) × genotype) on cognitive performance measures are presented in Table 1. The main effect of diagnosis revealed that schizophrenia subjects scored significantly lower in all cognitive assessments than the controls (*p* < 0.001). When all genotypes were examined (TT:TC:CC), there was a main effect of genotype for WASI only (*p* = 0.036), and post hoc testing revealed that TT subjects scored better on the WASI test than TC subjects (*p* = 0.048). Additionally, there were significant diagnosis × genotype interactions for WASI in all subjects (*p* = 0.018) and in male subjects (*p* = 0.009). When allele carriers were examined (TT:TC/CC), there was a significant main effect of allele carrier status in all subjects for RBANS total score (*p* = 0.035), RBANS constructional memory (*p* = 0.044), and RBANS language (*p* = 0.041), irrespective of diagnosis. Post hoc analyses revealed that, in all cases, the minor allele carriers (TC/CC) scored lower on these tests than the major allele homozygotes (0.035 ≤ *p* ≤ 0.049). Correlations between methylation status and cognitive measures (0.007 ≤ *p* ≤ 0.048) can be found in Appendix A.

**Table 1 ijms-24-15624-t001:** Results of two-way MANOVA performed to determine diagnosis (control or schizophrenia), genotype (or allele carrier), and diagnosis × genotype effects on cognitive performance measures, in all subjects (both sexes) and male and female subjects individually. Note that least significant difference (LSD) post hoc tests were for the results of significant diagnosis × genotype interactions only (highlighted in bold). Column a indicates the analysis considering the three genotypes of rs3144 analysed independently, in relation to diagnosis and their effects on neuropsychological testing. Column b indicates the analysis considering the alleles carriers (major genotype TT for rs3144 against minor allele carriers, i.e., both TC and CC genotypes together, in relation to diagnosis and their effects on neuropsychological testing).

		(a) Lingo-1 rs3144 (TT:TC:CC)			(b) Lingo-1 rs3144 (TT:TC/CC)		
		All	Male	Female	All	Male	Female
WTAR	Diagnosis	**F = 39.331, *p* < 0.001**	**F = 23.760, *p* < 0.001**	**F = 12.561, *p* = 0.001**	**F = 40.358, *p* < 0.001**	**F = 28.109, *p* < 0.001**	**F = 10.161, *p* = 0.002**
	Genotype	F = 31.953, *p* = 0.144	F = 2.865, *p* = 0.059	F = 0.396, *p* = 0.674	F = 2.154, *p* = 0.143	F = 2.710, *p* = 0.101	F = 0.001, *p* = 0.991
	Diagnosis × Genotype	F = 2.532, *p* = 0.081	F = 2.832, *p* = 0.061	F = 0.914, *p* = 0.404	F = 3.097, *p* = 0.079	F = 1.726, *p* = 0.190	F = 0.863, *p* = 0.355
WASI	Diagnosis	**F = 112.981, *p* < 0.001**	**F = 72.539, *p* < 0.001**	**F = 39.214, *p* < 0.001**	**F = 116.248, *p* < 0.001**	**F = 81.883, *p* < 0.001**	**F = 34.071, *p* < 0.001**
	Genotype	**F = 3.347, *p* = 0.036**	F = 1.847, *p* = 0.160	F = 1.536, *p* = 0.220	F = 1.122, *p* = 0.290	F = 0.040, *p* = 0.842	F = 2.532, *p* = 0.115
	Diagnosis × Genotype	**F = 4.070, *p* = 0.018**	**F = 4.801, *p* = 0.009**	F = 1.084, *p* = 0.342	**F = 4.512, *p* = 0.034**	F = 3.305, *p* = 0.070	F = 1.369, *p* = 0.245
	*LSD post-hoc*	HC:CC>TC, *p* = 0.638; HC:TT>TC, *p* = 0.459	HC:TC>CC, *p* = 0.675; HC:TC>TT, *p* = 0.125	-	HC:TC/CC>TT, *p* = 0.320	-	-
		**SZ:TT>TC, *p* = 0.025;** **SZ:CC>TC, *p* = 0.028**	**SZ:CC>TC, *p* = 0.015;** SZ:TT>TC, *p* = 0.083	-	SZ:TT>TC/CC, *p* = 0.137	-	-
LNS	Diagnosis	**F = 66.889, *p* < 0.001**	**F = 58.296, *p* < 0.001**	**F = 9.298, *p* = 0.003**	**F = 82.161, *p* < 0.001**	**F = 71.192, *p* < 0.001**	**F = 12.672, *p* = 0.001**
	Genotype	F = 0.159, *p* = 0.853	F = 0.202, *p* = 0.817	F = 0.218, *p* = 0.804	F = 0.143, *p* = 0.705	F = 0.012, *p* = 0.913	F = 0.014, *p* = 0.908
	Diagnosis x Genotype	F = 1.493, *p* = 0.226	F = 2.109, *p* = 0.124	F = 0.672, *p* = 0.513	F = 0.034, *p* = 0.853	F = 0.722, *p* = 0.396	F = 1.233, *p* = 0.270
COWAT	Diagnosis	**F = 42.645, *p* < 0.001**	**F = 33.011, *p* < 0.001**	**F = 7.251, *p* = 0.008**	**F = 40.637, *p* < 0.001**	**F = 35.413, *p* < 0.001**	**F = 5.082, *p* = 0.026**
	Genotype	F = 1.001, *p* = 0.369	F = 0.615, *p* = 0.542	F = 0.232, *p* = 0.794	F = 0.639, *p* = 0.423	F = 0.114, *p* = 0.736	F = 0.519, *p* = 0.473
	Diagnosis × Genotype	F = 2.450, *p* = 0.088	F = 1.809, *p* = 0.166	F = 0.915, *p* = 0.404	**F = 4.062, *p* = 0.045**	F = 2.191, *p* = 0.140	F = 1.496, *p* = 0.224
	*LSD post-hoc*	-	-	-	HC:TC/CC>TT, *p* = 0.316	-	-
		-	-	-	SZ:TT>TC/CC, *p* = 0.190	-	-
RBANS TOTAL	Diagnosis	**F = 187.869, *p* < 0.001**	**F = 139.835, *p* < 0.001**	**F = 43.442, *p* < 0.001**	**F = 215.992, *p* < 0.001**	**F = 166.555, *p* < 0.001**	**F = 47.114, *p* < 0.001**
	Genotype	F = 2.486, *p* = 0.085	F = 1.757, *p* = 0.175	F = 2.236, *p* = 0.112	**F = 4.478, *p* = 0.035**	F = 2.119, *p* = 0.147	F = 1.694, *p* = 0.196
	Diagnosis × Genotype	F = 2.887, *p* = 0.057	**F = 3.276, *p* = 0.040**	F = 0.018, *p* = 0.982	F = 1.015, *p* = 0.315	F = 0.869, *p* = 0.352	F = 0.022, *p* = 0.882
	*LSD post-hoc*	-	HC: TC>CC, *p* = 0.365; HC:TC>TT, *p* = 0.970	-	-	-	-
		-	**SZ: CC>TC, *p* = 0.015;** **SZ: TT>TC, *p* = 0.025**	-	-	-	-
RBANS delayed memory	Diagnosis	**F = 91.653, *p* < 0.001**	**F = 64.576, *p* < 0.001**	**F = 23.251, *p* < 0.001**	**F = 109.352, *p* < 0.001**	**F = 79.857, *p* < 0.001**	**F = 26.056, *p* < 0.001**
	Genotype	F = 0.591, *p* = 0.554	F = 0.396, *p* = 0.673	F = 3.072, *p* = 0.051	F = 0.952, *p* = 0.330	F = 0.244, *p* = 0.622	F = 0.480, *p* = 0.490
	Diagnosis × Genotype	F = 1.533, *p* = 0.217	F = 1.914, *p* = 0.150	F = 0.374, *p* = 0.689	F = 0.215, *p* = 0.643	F = 0.358, *p* = 0.551	F = 0.201, *p* = 0.655
RBANS attention	Diagnosis	**F = 121.514, *p* < 0.001**	**F = 102.069, *p* < 0.001**	**F = 18.051, *p* < 0.001**	**F = 133.575, *p* < 0.001**	**F = 120.552, *p* < 0.001**	**F = 17.471, *p* < 0.001**
	Genotype	F = 1.375, *p* = 0.254	F = 0.823, *p* = 0.440	F = 0.491, *p* = 0.613	F = 2.424, *p* = 0.120	F = 1.223, *p* = 0.270	F = 0.598, *p* = 0.441
	Diagnosis × Genotype	F = 1.999, *p* = 0.137	F = 1.773, *p* = 0.172	F = 0.295, *p* = 0.745	F = 1.898, *p* = 0.169	F = 0.765, *p* = 0.383	F = 0.582, *p* = 0.447
RBANS immediate memory	Diagnosis	**F = 109.252, *p* < 0.001**	**F = 70.728, *p* < 0.001**	**F = 32.401, *p* < 0.001**	**F = 132.338, *p* < 0.001**	**F = 95.152, *p* < 0.001**	**F = 33.525, *p* < 0.001**
	Genotype	F = 0.817, *p* = 0.443	F = 0.157, *p* = 0.855	F = 2.182, *p* = 0.118	F = 0.509, *p* = 0.476	F = 0.158, *p* = 0.691	F = 0.277, *p* = 0.600
	Diagnosis × Genotype	F = 1.172, *p* = 0.311	F = 2.411, *p* = 0.092	F = 0.348, *p* = 0.707	F = 0.001, *p* = 0.988	F = 0.028, *p* = 0.868	F = 0.001, *p* = 0.970
RBANS constructional	Diagnosis	**F = 54.969, *p* < 0.001**	**F = 44.151, *p* < 0.001**	**F = 15.657, *p* < 0.001**	**F = 61.592, *p* < 0.001**	**F = 47.775, *p* < 0.001**	**F = 17.768, *p* < 0.001**
	Genotype	F = 2.699, *p* = 0.069	F = 3.139, *p* = 0.045	F = 0.659, *p* = 0.520	**F = 4.071, *p* = 0.044**	F = 3.198, *p* = 0.075	F = 1.012, *p* = 0.317
	Diagnosis × Genotype	F = 2.786, *p* = 0.063	F = 2.290, *p* = 0.104	F = 0.460, *p* = 0.638	F = 2.005, *p* = 0.158	F = 2.317, *p* = 0.129	F = 0.274, *p* = 0.602
RBANS language	Diagnosis	**F = 58.008, *p* < 0.001**	**F = 32.662, *p* < 0.001**	**F = 23.004, *p* < 0.001**	**F = 60.627, *p* < 0.001**	**F = 35.397, *p* < 0.001**	**F = 24.245, *p* < 0.001**
	Genotype	F = 2.453, *p* = 0.088	F = 1.104, *p* = 0.333	F = 1.980, *p* = 0.144	**F = 4.223, *p* = 0.041**	F = 1.177, *p* = 0.279	**F = 3.970, *p* = 0.049**
	Diagnosis × Genotype	F = 0.286, *p* = 0.751	F = 0.271, *p* = 0.763	F = 0.064, *p* = 0.938	F = 0.473, *p* = 0.492	F = 0.338, *p* = 0.562	F = 0.023, *p* = 0.880

Abbreviations: CD, schizophrenia subjects classified as having generalized cognitive deficit; CI, confidence interval; CS, schizophrenia subjects classified as cognitively spared; HC, healthy controls; CC, minor homozygote genotype; COWAT, Controlled Oral Word Association Test; HC, healthy controls; LNS, Letter Number Sequencing Test; LSD, least significant difference; RBANS, Repeatable Battery for the Assessment of Neuropsychological Status; TC, heterozygote genotype; TT, major homozygote genotype; TC/CC, minor allele (C) Carrier; SZ, schizophrenia subjects; WASI, Wechsler Abbreviated Scale of Intelligence; WTAR, Wechsler Test for Adult Reading.

The regression parameters and models implemented in the multinomial logistic regressions, used to investigate the ability of rs3144 to predict membership of subjects in the grade of membership (GoM) classifications of being cognitively spared or as having cognitive deficit, can be found in Table 2. In Model 1, the dependent variables were the GoM subtypes, with controls being dummy coded as the reference variable and the genotype or allele carrier status of rs3144 being the fixed factor. In this model, the heterozygote TC genotype was predictive of membership to the CD subtype in all subjects (*p* = 0.047) and in males (*p* = 0.034); however, this only reached model significance in males (*p* = 0.010). In Model 1, allele carrier status failed to predict the membership of CS or CD subtypes in reference to control status.

In Model 2, the ability of genotype or allele carriers to predict membership within schizophrenia subjects to the CD subtype with the CS subtype as the reference category was examined. Again, the heterozygous TC genotype was predictive of membership to the CD subtype in all schizophrenia subjects (*p* = 0.040) with a model significance of *p* = 0.044. Males reached borderline significance, with the TC genotype being more highly represented in the CD subtype (*p* = 0.051), despite reaching overall model significance (*p* = 0.007).

One-way ANOVA results for differences in methylation status between controls and schizophrenia subjects, divided into CS and CD subtypes, can be found in Appendix A. The results showed that only methylation status at CpG5 and CpG10 was affected by these subtypes (*p* ≤ 0.050). Post hoc analyses revealed that, in both cases, methylation at these sites was higher in controls compared to the CD subtype (*p* ≤ 0.023); additionally, controls also displayed higher methylation than the CS subtype at CpG10 (*p* = 0.023).

**Table 2 ijms-24-15624-t002:** Summary of logistic regressions for predicting membership of CD subtype on the basis of genotype or allele carrier classification for Model 1 (healthy control used as reference) and for Model 2 (within schizophrenia group only, cognitively spared schizophrenic used as reference). Significant results are highlighted in bold.

SNP (Risk Allele Carrier)	Model Significance			Predictor				
			Pseudo R^2^ (Cox and Snell)	B Value	Standard Error	*p* Value	(95% CI)	Exp (B)
rs3144 (genotype)	Model 1 (reference category = HC)		Predicting CS subtype:					
	all	χ^2^ = 9.143, d.f. = 4, *p* = 0.058	0.028	−0.177	0.324	0.583	(0.444–1.579)	0.837
	male	**χ^2^ = 13.389, d.f. = 4, *p* = 0.010**	0.059	−0.123	0.396	0.756	(0.407–1.923)	0.884
	female	χ^2^ = 2.320, d.f. = 4, *p* = 0.677	0.022	−0.285	0.561	0.611	(0.251–2.256)	0.752
			Predicting CD subtype:					
			**0.028**	**0.651**	**0.328**	**0.047**	**(1.008–3.648)**	**1.917**
			0.059	**0.806**	**0.381**	**0.034**	**(1.062–4.725)**	**2.24**
			0.022	0.071	0.687	0.917	(0.280–4.126)	1.074
	Model 2 (reference category = CS type)		Predicting CD subtype					
	all	**χ^2^ = 6.257, d.f. = 2, *p* = 0.044**	**0.047**	**0.828**	**0.404**	**0.040**	**(1.038–5.050)**	**2.289**
	male	**χ^2^ = 9.944, d.f. = 2, *p* = 0.007**	0.098	0.930	0.476	0.051	(0.997–6.438)	2.533
	female	χ^2^ = 1.822, d.f. = 2, *p* = 0.402	0.051	0.357	0.802	0.656	(0.297–6.877)	1.429
rs3144 (minor allele carrier)	Model 1 (reference category = HC)		Predicting CS subtype:					
	all	χ^2^ = 3.823, d.f. = 2, *p* = 0.148	0.012	0.101	0.284	0.723	(0.634–1.930)	1.106
	male	χ^2^ = 3.659, d.f. = 2, *p* = 0.160	0.017	0.277	0.345	0.421	(0.672–2.593)	1.32
	female	χ^2^ = 0.983, d.f. = 2, *p* = 0.612	0.009	−0.296	0.51	0.561	(0.274–2.021)	0.744
			Predicting CD subtype:					
			0.012	0.596	0.312	0.056	(0.985–3.342)	1.815
			0.017	0.678	0.367	0.064	(0.960–4.043)	1.97
			0.009	0.397	0.6	0.437	(0.459–4.821)	1.487
	Model 2 (reference category = CS type)		*Predicting CD subtype*					
	all	χ^2^ = 1.833, d.f. = 1, *p* = 0.176	0.014	0.495	0.368	0.178	(0.798–3.375)	1.641
	*male*	χ^2^ = 0.862, d.f. = 1, *p* = 0.353	0.009	0.401	0.433	0.355	(0.639–3.490)	1.493
	*female*	χ^2^ = 0.982, d.f. = 1, *p* = 0.322	0.028	0.693	0.707	0.327	(0.500–7.997)	2.000

Abbreviations: CD, cognitive deficits; CS, cognitively spared; HC, healthy control; SNP, single nucleotide polymorphism.

### 2.3. Rs3144 and Methylation Effects on FA

Two-way (diagnosis (or allele carrier) × genotype) ANOVA showed a main effect of diagnosis on FA levels (*p* ≤ 0.010), in all cases, and males and females separately. FA levels were higher in controls compared to in schizophrenia subjects (Table 3). There was a significant main effect of genotype in male subjects only (*p* = 0.018), with post hoc tests showing that FA levels were higher in those subjects with the CC genotype than TC genotype (*p* = 0.012). Significant correlations were found between FA measures and methylation status in male and female controls at CpG6 and CpG9, respectively (Appendix A).

## 3. Materials and Methods

### 3.1. Participants

The participants for this study were recruited as a part of an ongoing study to collect a large national repository of patient samples and associated data, known as the Australian Schizophrenia Research Bank (ASRB) [20]. All subjects recruited provided written informed consent for analysis of their data. Subjects with schizophrenia were identified using the Diagnostic and Statistical Manual of Mental Disorders-IV criteria [20]. All subjects recruited were subjected to thorough psychometric testing (see Appendix A). This study was approved by and conducted in accordance with the University of Wollongong Human Research Ethics Committee (HE10/161) and the University of New South Wales Human Ethics Committee (HC12658) guidelines.

### 3.2. Participants Selection Criteria

The subjects selected for inclusion in the present study were chosen and matched using strict selection criteria, to prevent population stratification. Cases were matched with controls (no prior personal or family history of mental disorders) according to sex and age. The complete sample consisted of 268 schizophrenia cases (186 males and 82 females, with an average age of 38.94 ± 10.99 years) and 268 matched controls (169 males and 99 females, with an average age of 38.60 ± 12.57 years). The ethnic origin of the Caucasian volunteers was determined using the participant’s response to questions regarding their birth place, in addition to that of their family members 2 generations before them on both maternal and paternal sides of the family. Of the 536 subjects in this study, 458 (85.4%) were born in Australia or New Zealand, 43 (8.0%) were born in the United Kingdom, and 12 (2.2%) were born in the United States or Canada; the remaining 23 (4.4%) were born in Europe. All subjects were of (self-reported) European descent. The majority of schizophrenia subjects were medicated: 58 patients were taking typical antipsychotic drugs, 211 were taking atypical antipsychotic drugs, 54 were on mood stabilizing drugs, and 92 patients were taking antidepressants.

### 3.3. Neuropsychological Measures and MRI Acquisition

Premorbid and current intelligence quotient (IQ) were obtained using the Wechsler Test for Adult Reading (WTAR) [21] and Wechsler Abbreviated Scale of Intelligence (WASI) [22], respectively. The Controlled Oral Word Association Test (COWAT) [23] and the Letter Number Sequencing Test (LNS) [22] were used to assess levels of executive function and working memory. Finally, indices of attention and memory function, specifically delayed memory, immediate memory, constructional memory, language, and attention were derived from The Repeatable Battery for the Assessment of Neuropsychological Status (RBANS) [24]. A large subset of the schizophrenia subjects included in this study were also classified as either cognitively spared (CS) or as having a generalized cognitive deficit (CD) by a previous study using an ASRB cohort [25]. A large subset of this cohort also underwent structural magnetic resonance imaging brain scans, to obtain diffusion weighted images as an index of WM integrity (see Appendix A).

### 3.4. Genotyping and Methylation Analyses

Rs3144 within the *Lingo-1* 3′-UTR was selected for analyses in our schizophrenia case-control population based on its minor allele frequency (MAF), with C being the minor allele reported in Caucasian populations (Table 4), and its previous association with other neurological disorders [2]. SNP genotyping was performed using a MassARRAY^®^ genotyping assay (Sequenom, Inc., San Diego, CA, USA), with the analysis performed through matrix-assisted laser desorption/ionization time-of-flight mass spectrometry (MALDI-TOF MS). Due to the high density of CpG sites surrounding rs3144 (Figure 1a), Sequenom methylation analysis was used to analyse the percentage of methylation at CpG sites (CpG1-CpG10) within the *Lingo-1* 3′-UTR (see Appendix A).

### 3.5. Statistical Methods

Analyses were performed using SPSS (version 22.0, SPSS Inc., Chicago, IL, USA), with the significance for all statistical tests set to *p* ≤ 0.05. All analyses were initially performed on all subjects (i.e., both sexes combined), and then by gender. The genotypic distribution of rs3144 was assessed for deviation from Hardy–Weinberg equilibrium (HWE). Chi-square analyses examined differences in allele (T:C) and genotype (TT:TC:CC) frequencies for rs3144 between cases and controls. Given the low frequency of minor homozygote genotypes in the tested cohort, subjects classified as minor allele carriers (TC/CC) were compared to major allele homozygotes (TT). Data are expressed as specific counts for alleles and genotypes. Methylation data were inverse-normal transformed (Blom’s transformation) to give a near normal distribution. Two-way multivariate analyses of variance (MANOVA) were performed, to determine the effects of diagnosis (control/schizophrenia), genotype (or allele carrier status), or interactions between diagnosis and genotype (or carrier) on (i) each of the 10 tested CpG sites and (ii) each of the 10 cognitive performance measures. Two-way analyses of variance (ANOVA) were performed to determine diagnosis, genotype, and diagnosis × genotype effects on FA status. Significant interactions were followed up with one-way ANOVA with least significant difference (LSD) post hoc tests to identify genotype effects on methylation status, cognitive performance measures, or FA status within each diagnostic group. One-way ANOVA was also performed to determine differences in methylation status between controls and schizophrenia subjects classified into CS or CD subtypes. The capacity for genotypes or allele carriers to predict membership to either CS or CD subtypes was examined using multinomial logistic regressions, with CS and CD subtypes as the dependent variables. The first series of analyses used controls as the reference category, to determine whether a certain genotype or allele carrier status was more prominent in the CS or CD subtype. The second series of analyses used the CS subtype as the reference category, to assess if a certain genotype or allele carrier status was over-represented in the CD subtype. Pearson’s correlations determined interactions between methylation status and (i) cognitive performance measures or (ii) FA in all subjects; controls, schizophrenia subjects, and male and female control and schizophrenia subjects were considered separately.

## 4. Discussion

For decades, a vast number of genes and genetic markers, including SNPs, have been involved in the genetic susceptibility of schizophrenia (up to 80%) [26,27]. While SNPs contribute in a somewhat limited manner [28], their cumulative effect could be responsible for up to 30% of the genetic heritability of schizophrenia [29]. Thus, novel SNPs identified as a risk factor for schizophrenia should be considered with interest. Here, we identified, for the first time, the association of *Lingo-1* genetic marker (rs3144) with schizophrenia in a case-control cohort. Major rs3144 TT genotype schizophrenia subjects were found to have significantly higher scores in executive and cognitive function, as determined by WASI. Considering the ever growing evidence reporting a significant role of environmental risk factors acting in parallel with genetic risk factors and contributing to schizophrenia susceptibility [30,31], we examined the methylation status of CpG sites within the vicinity of rs3144 across the 3′-UTR. Schizophrenia subjects had a significantly lower percentage of methylation at CpG5 (immediately downstream of rs3144) in CD subtypes compared to controls. Since differential methylation in the 3′-UTR has been shown to be negatively correlated with gene and protein expression [12,32,33], the results of the present study suggest a synergistic effect of both genotype and methylation status in schizophrenia. This synergic effect may cause an upregulation of protein expression in the brains of schizophrenia patients, as was observed in our previous postmortem human study (Figure 1b (i),c (i)) [7].

Furthermore, when taking into account that disruption of WM integrity in schizophrenia is one of the hallmarks of this disorder and considering the major role of Lingo-1 in the regulation of myelination [34], we examined whole-brain FA levels and found a significant decrease in schizophrenia subjects compared to controls, in accordance with the literature [35]. Interestingly, males with the CC minor genotype have shown a higher level of FA compared to T allele carriers in schizophrenia, suggesting a potential impact of *Lingo-1* on brain myelination. Gender specific differences in WM integrity were also reported in a larger case-control cohort for schizophrenia (1963 cases vs. 2359 controls), with females exhibiting significantly lower brain FA overall compared to males [36]. WM abnormalities were also found to be more pronounced in female patients compared to males in schizophrenia cohorts through DTI and MRI analysis [37,38] and postmortem studies [39,40]. Furthermore, gender differences were also observed in healthy cohorts, with men showing a larger white matter volume on average compared to women [41,42,43]. Although, myelination increases in a linear manner from infants to middle age adulthood, the trajectory of age-related WM volume expansion for schizophrenia patients deviates from that observed for age-matched normal controls [41,44]. Taking into account that males demonstrate more severe schizophrenia symptoms than females and considering differential expression of rs3144 genotypes by gender, it is reasonable to hypothesize a role for Lingo-1 in gender-specific-related WM levels and schizophrenia symptoms.

It is also interesting to note that that myelination processes from late adolescence to early adulthood (the time of peak incidence for schizophrenia onset) are more pronounced in the frontal and temporal lobes, brain regions critical in the pathophysiology of schizophrenia [8,44]. Our previous work investigating the expression profile of Lingo-1 in a male pharmacological model of schizophrenia, reported a significant increase in Lingo-1 expression in the frontal cortex in adolescent and young adult animals compared to neonate rats [45]. Future work on Lingo-1 expression according to age and genotypes for rs3144 would be necessary to further examine this finding in men and women.

Additionally, both genotype and methylation levels in the 3′-UTR were found to have some associations with FA measures. Altogether, this is the first study to simultaneously report the effects of genetic and epigenetic profiles in the *Lingo-1* 3′ flanking region in schizophrenia, and that genetic and epigenetic alterations in the 3′-UTR may be associated with decreased cognitive performance and WM integrity in schizophrenia.

We are the first to study *Lingo-1* in the context of schizophrenia, reporting a significant genetic association with this disorder in a limited but well-characterized cohort. However, this finding should be replicated in an independent and larger case-control population for schizophrenia. The odds of having schizophrenia were found to be 40% higher in the general population for individuals with the CC genotype compared to other genotypes for rs3144. Since the present study showed a differential MAF for rs3144 between schizophrenia and control subjects, and considering our previous finding showing a higher level of Lingo-1 protein in the DLPFC, we suggest that the minor allele carrier genotype may lead to differential gene expression, consequently resulting in higher levels of Lingo-1 protein in the schizophrenia brain (Figure 1c (i)) [7]. The mechanism underlying this process may be partly due to differential levels of microRNA, which can modulate transcription in the 3′-UTR by silencing mRNA and/or to regulating gene expression at the post-transcriptional level [46]. In support of this hypothesis, a study examining microRNA expression in the prefrontal cortex found a number of downregulated microRNAs in schizophrenia brains compared to controls [47]. Interestingly, one of these downregulated microRNA, hsa-miR-26b, has been shown to target the *Lingo-1* 3′-UTR [48]. Moreover, SNPs in the 3′-UTR were shown to affect the function of microRNAs [49]. Since Lingo-1 is a known negative regulator of the neuronal processes underlying synaptic plasticity, it is reasonable to suggest that the minor allele for rs3144 may induce differential expression of microRNAs in the *Lingo-1* 3′-UTR, leading to an increase in Lingo-1 protein expression, as previously observed in schizophrenia (Figure 1c) compared to healthy controls (Figure 1b) [7].

Differentially methylated genes have been previously reported in postmortem tissues, blood, and saliva from schizophrenia sufferers compared to controls, with a higher density of methylated CpG sites in the tested gene and/or the full genome for the controls compared to schizophrenia groups [50,51,52]. As illustrated in Figure 1b,c, differential DNA methylation levels can directly influence the levels of protein expression. In eukaryotes, DNA methylation is a biochemical process with the ability to “switch off” gene expression when required, allowing for proper functioning of the cell [53]. If a higher percentage of DNA methylation is observed globally across the full genome, as proposed in Figure 1b (ii), and within the *Lingo-1* 3′-UTR (Figure 1b (i)) for healthy controls, as observed in the present study, we suggest that the synergic effect of both methylation processes is suppressing the expression of *Lingo-1* gene and subsequently protein levels in controls. This collaborative effect would result in maintaining Lingo-1 protein expression at an appropriate level for the proper functioning of cells, allowing healthy neuronal growth and survival (Figure 1b (i) and (ii)).

In contrast, we hypothesize that global hypomethylation across the genome (Figure 1c (ii)) and specifically at the *Lingo-1* 3′-UTR (Figure 1c (i)), as shown in the present study, interacts with the minor allele present in the 3′-UTR (and/or microRNA as discussed above) in schizophrenia patients, inhibiting the “switch off” effect normally induced by methylation, consequently resulting in a higher level of protein expression.

When considering the ability of rs3144 to predict membership to GoM classified CS or generalized CD subtypes of schizophrenia, we found that rs3144 was able to significantly predict cognitive impairment in both statistical models. This outcome indicates that rs3144 has the ability to predict cognitive dysfunction in schizophrenia in addition to the general population. Considering the key role of Lingo-1 in numerous neuronal processes that critically underlie learning and memory processes, the results of our genetic analysis in relation to cognitive dysfunction remain consistent. Significant negative correlations were reported between methylation and neuropsychological measures in schizophrenia, and in male schizophrenia subjects, for LNS (a measure of executive function and working memory), total RBANS score, and RBANS delayed memory with the CpG4 site adjacent to rs3144. In this context, we hypothesize that the effects of hypermethylation at CpG4 may have been overcome by the increased frequency of the minor alleles expressed in the schizophrenia group, resulting in a higher level of Lingo-1 protein expression, and consequently altering higher order cognitive functions, which involve both the prefrontal cortex and hippocampal brain regions.

In an attempt to test the above mentioned hypothesis, we genotyped for rs3144 using DNA from the CA1 and CA3 hippocampal regions in an independent case-control cohort (n = 40) [7]. Due to the limited number of samples in this cohort, the genotype for rs3144 was only analysed in relation to Lingo-1 protein levels (and not in relation to diagnosis), to assess a potential functional role for this SNP. As illustrated in Appendix A, the major TT genotype for rs3144 was associated with higher levels of Lingo-1 observed in the CA1 (*p* = 0.022), while the same major genotype seems to have the complete opposite effect in the CA3, although not reaching significance. Interestingly, global levels of DNA methylation in the rat brain were reported to be significantly higher in CA1 compared to CA3 regions [54]. Considering that increased DNA methylation is essential for memory consolidation processes [55] and taking into account the specific implication of the CA1 in memory consolidation [55], we suggest that the negative effects of elevated Lingo-1 protein levels in relation to the major TT genotype may be counteracted by the positive effects of methylation, stimulating synaptic plasticity mechanisms, which critically underlie memory consolidation. In addition, and in support of this hypothesis, the methylation status at CpG5, adjacent to rs3144, was shown to have a significant association with cognitive subtype, with controls being hypermethylated at this site compared to schizophrenia subtypes with generalized cognitive deficit, supporting the idea of a collaborative effect of methylation and genotype on cognitive performance and memory consolidation.

FA has been found to be significantly lower in schizophrenia subjects compared to controls but also in schizophrenia subjects with cognitive impairment [4,56]. In this study, we found significant positive correlations between the scores of all the neuropsychological tests performed in our cohort (except for WTAR) with the level of whole-brain FA (*p* < 0.022). A significant effect of genotype on FA measures was found in males, with post hoc analysis reporting that minor CC subjects had increased FA levels compared to TC subjects, supporting a potential functional role of this SNP in regulating WM integrity. Methylation levels in the vicinity of rs3144 (CpG6) and further down on the 3′-UTR (CpG9) were significantly correlated with FA levels in healthy males and females, respectively, suggesting that methylation in the *Lingo-1* 3′-UTR, in synergy with global genome methylation, which is known to be higher in males than in females [57], may have a protective effect on WM integrity (Figure 1b). Since antipsychotics can affect DNA methylation [47,57,58], the effects of medication on the levels of DNA methylation were examined in this study and found to be non-significant (*p* > 0.05). Altogether, we hypothesize that CC genotype carriers for rs3144 *Lingo-1,* along with local and potentially global gene hypomethylation, may cause impaired WM integrity and neuronal growth. Both of these latter processes underlie cognitive function and/or dysfunction, as illustrated in Figure 1b,c. Determining both the genetic and epigenetic profiling for rs3144 and the surrounding sites may inform potential therapeutic actions to reduce Lingo-1-induced effects in the brain [59]. Future work will be necessary to validate this new avenue of therapy.

In summary, this study reports a potential involvement of *Lingo-1* at a genetic and epigenetic level in schizophrenia vulnerability. Both genetic and epigenetic risk factors have synergistic effects on intelligence and cognitive performance, but also on whole-brain WM heath. Although further investigation within the promoter and intragenic regions of *Lingo-1* will be necessary, *Lingo-1* may play a critical role in schizophrenia pathogenesis.

## Figures and Tables

**Figure 1 ijms-24-15624-f001:**
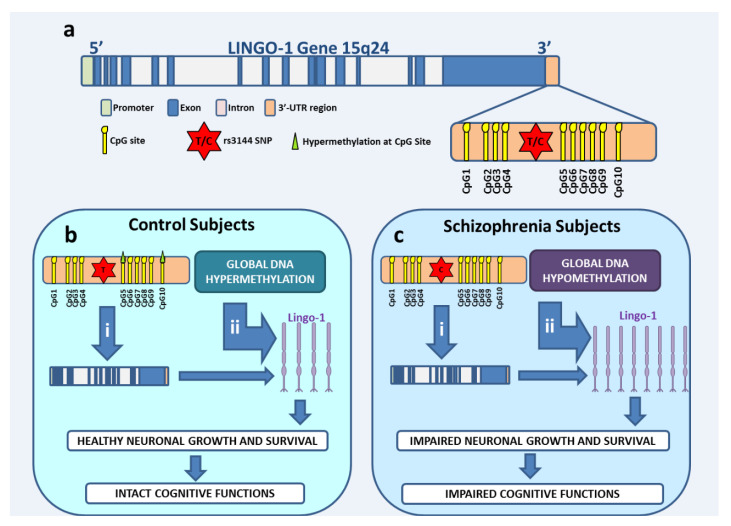
Schematic representation of the *Lingo-1* gene and hypothesized effects of genetic and epigenetic alterations in the *Lingo-1* 3′-UTR regulatory region. (**a**) Schematic representation of the *Lingo-1* gene with the 3′-UTR magnified and the rs3144 SNP and surrounding CpG sites identified and tested in the study. (**b**) Healthy controls potential mechanism of action for *Lingo-1*: hypothesized effect of major T allele and hypermethylation occurring both in (i) the 3′-UTR of *Lingo-1*, and (ii) globally across the genome, respectively, in healthy control subjects resulting in the maintenance of moderate levels of Lingo-1 protein in the brain (represented in purple). Appropriate levels of Lingo-1 protein lead to healthy neuronal growth and survival and subsequently intact cognitive functioning and WM integrity. (**c**) Potential mechanism of action for *Lingo-1* in schizophrenia: hypothesized effect of the minor C allele and hypomethylation occurring both in (i) the 3′-UTR of *Lingo-1* and (ii) globally across the genome, respectively, in schizophrenia subjects, causing an increase in Lingo-1 protein expression in the brain (represented in purple). Increased levels of Lingo-1 protein in the brain lead to impaired neuronal growth and survival and subsequently impaired cognitive function and decreased WM integrity.

**Table 3 ijms-24-15624-t003:** Results of two-way ANOVA performed to determine diagnosis, genotype, and diagnosis × genotype effects on fractional anisotropy status, in all subjects (both sexes) and male and female subjects individually. Note that least significant difference (LSD) post hoc tests are shown only for the significant results highlighted in bold. Column a indicates the analysis considering the three genotypes of rs3144 analysed independently in relation to diagnosis and their effects on neuropsychological testing. Column b indicates the analysis considering the alleles carriers (major genotype TT for rs3144 against minor allele carriers, i.e., both TC and CC genotypes together in relation to diagnosis and their effects on neuropsychological testing).

		(a) Lingo-1 rs3144 (TT:TC:CC)			(b) Lingo-1 rs3144 (TT:TC/CC)		
		All	Male	Female	All	Male	Female
FA	Diagnosis	**F = 12.840, *p* < 0.001**	**F = 5.794, *p* = 0.018**	**F = 9.181, *p* = 0.004**	**F = 13.482, *p* < 0.001**	**F = 6.921, *p* = 0.010**	**F = 8.406, *p* = 0.006**
	Genotype	F = 2.158, *p* = 0.119	**F = 3.368, *p* = 0.038**	F = 0.071, *p* = 0.932	F = 0.152, *p* = 0.697	F = 0.066, *p* = 0.798	F = 0.071, *p* = 0.792
	Diagnosis × Genotype	F = 0.066, *p* = 0.937	F = 0.146, *p* = 0.864	F = 0.672, *p* = 0.517	F = 0.286, *p* = 0.593	F = 0.141, *p* = 0.708	F = 0.432, *p* = 0.515
	LSD post-hoc	**HC > SZ, *p* < 0.001**	**HC > SZ, *p* = 0.018**	**HC > SZ, *p* = 0.004**	**HC > SZ, *p* < 0.001**	**HC > SZ, *p* = 0.010**	**HC > SZ, *p* = 0.006**
		-	**CC > TC, *p* = 0.012**	-	-	-	-

Abbreviations: FA: fractional anisotropy; SZ, schizophrenic sufferers; HC, healthy controls, TC, heterozygote genotype; TT, major homozygote genotype; CC, minor homozygote genotype; LSD, least significant difference.

**Table 4 ijms-24-15624-t004:** Rs3144 SNP in the *Lingo-1* 3’-UTR and its minor allelic and genotypic frequencies and tests of allelic association in the tested cohort. The table below provides information on the allelic frequencies (column a), allele carriers (column b, meaning minor allelic carriers (CC and TC) vs. TT, and genotypic frequencies (column c: three genotypes analysed independently).

		a MAF (%)				Allelic Association		b MCF (%)				Allele Carrier Association		c MGF (%)				Genotypic Association		Odds Ratio (OR)	95% CI	*p*
						(HC:SZ only)						(HC:SZ only)						(HC:SZ only)				
		HC	SZ	CS	CD	χ^2^	p	HC	SZ	CS	CD	χ^2^	p	HC	SZ	CS	CD	χ^2^	p			
Rs 3144	all	35.64	43.31	42.03	42.74	4.28	**0.03**	56.12	65.49	57.97	69.35	3.01	0.08	14.80	21.13	31.88	53.23	3.88	0.14	1.39	(1.016–1.901)	**0.03**
	male	34.40	43.63	45.92	50.00	4.04	**0.04**	55.20	66.67	61.22	66.67	3.09	0.07	13.60	20.59	30.61	33.33	3.76	0.15	1.47	(1.008–2.158)	**0.04**
	**female**	**36.43**	**42.50**	**32.50**	**40.43**	**0.79**	**0.37**	**57.75**	**62.50**	**50.00**	**70.21**	**0.24**	**0.62**	**16.90**	**22.50**	**35.00**	**59.57**	**0.57**	**0.75**	**1.24**	**(0.709–2.170)**	**0.44**

Abbreviations: CD, schizophrenia subjects classified as having generalized cognitive deficit; CI, confidence interval; CS, schizophrenia subjects classified as cognitively spared; HC, healthy controls; SZ, schizophrenia subjects; MAF, minor allele carrier frequency (T:C); MCF, minor carrier frequency (TT:TC/CC); MGF, minor genotypic frequency (TT:TC:CC); HWE, Hardy–Weinberg equilibrium. Significant values are in bold.

## Data Availability

The data that support the findings of this study are not openly available to protect participant privacy.

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
