# Peer review of "Genetic and Epigenetic Regulation in Lingo-1: Effects on Cognitive Function and White Matter Microstructure in a Case-Control Study for Schizophrenia"

_ijms, 2023, doi:10.3390/ijms242115624_

Round 1

Reviewer 1 Report

The manuscript titled "Genetic and epigenetic regulation in Lingo-1: effects on cognitive function and white matter microstructure in a case-control study for schizophrenia" by Jessica Andrews et al. examined Lingo-1's genetic and epigenetic risk factors in schizophrenia and their potential association with cognitive dysfunction and white matter integrity. In a schizophrenia case-control cohort, the authors found that rs3144 minor allele's frequency is overrepresented in the schizophrenia population. Moreover, the rs3144 genotype predicted generalized cognitive deficits in a subclass of schizophrenia subjects. The study is interesting, and the methods and data support the authors' conclusions. Here are some minor observations:

The authors found a significant association between the allelic frequency of rs3144 and schizophrenia that remained in the male schizophrenia subjects after a sex-specific analysis. A significant effect of genotype on FA measures was also found in males. Yet, the authors barely address the implications of these sex-specific results. The manuscript would benefit from a more thorough discussion of these results, the potential role of Lingo-1 in the context of sex-specific mechanisms in schizophrenia pathophysiology, and the impact of their findings in informing therapeutic approaches for both sexes.

- In Line 77, please remove the superscript (1) after the word cognition

- Table 4 -> Lacks abbreviation definitions

- For consistency, please abbreviate Fraction Anisotropy in Table 4

- Figure 1 should appear when first mentioned (in the methods section)

- Figure legend (Figure 1) should be organized in the order in which the information that it depicts is described in the text

Author Response

Please see response attached. thanks for your help

Reviewer 2 Report

General Comments: The article titled "Genetic and Epigenetic Alterations in Lingo-1 (LRIG6) in Schizophrenia: Implications for Cognitive Function and White Matter Integrity" investigates the role of Leucine-rich repeat and immunoglobulin domain-containing 48 protein (Lingo-1) in schizophrenia, focusing on genetic (single nucleotide polymorphism, SNP) and epigenetic (methylation status at CpG sites) markers within the Lingo-1 gene. The study aims to explore associations between these markers and schizophrenia and their implications for cognitive deficits and white matter integrity. The study combines genetic and epigenetic analyses, cognitive assessments, and white matter integrity measures in a well-characterized case-control cohort.

Here are the main points and recommendations:

  1. Clarity and Organization: The article is well-organized, with transparent sections detailing the materials and methods, results, and discussion. The introduction provides adequate background information, and the study's objectives are well-defined. The research questions and hypotheses are articulated.

  2.  
  3. Strengths:

    • The study employs a comprehensive approach, combining genetic and epigenetic analyses with cognitive assessments and white matter integrity measures, providing a holistic view of potential associations with schizophrenia.
    • Including genetic and epigenetic analyses strengthens the investigation of Lingo-1's role in schizophrenia, as it considers both the genetic variation and potential regulatory mechanisms.
    • The study uses a well-characterized case-control cohort, providing a solid foundation for the research.
    •  
  4. Methodology:

    • The methods section describes the study participants, data collection, genotyping, and statistical analyses. This transparency is commendable.
    • The use of the Australian Schizophrenia Research Bank (ASRB) as a source of participants adds credibility to the study.
    • The statistical methods are appropriate for the research questions, and the results are presented clearly.
    • The inclusion of sex-specific analyses is important, as it acknowledges potential gender differences in schizophrenia and Lingo-1 associations.
    •  
  5. Results:

    • The results are presented clearly and concisely, with tables and figures aiding in the interpretation of findings.
    • The significant associations between Lingo-1 genetic variations, cognitive function, and white matter integrity are noteworthy.
    • Identifying potential epigenetic markers (methylation status at CpG sites) associated with schizophrenia and cognitive deficits adds depth to the study.
  6. Discussion:

    • The discussion section thoroughly interprets the results in the context of existing literature.
    • The discussion of the potential mechanisms underlying Lingo-1's involvement in schizophrenia, including genetic and epigenetic factors, is well-presented.
    • The implications of the findings for schizophrenia pathogenesis, cognitive deficits, and white matter integrity are discussed comprehensively.
    •  
  7. Limitations and Future Directions:

    • The study acknowledges certain limitations, such as the need for replication in larger cohorts and the potential influence of medication on DNA methylation. These limitations are appropriately addressed.
    • Future directions are suggested, including further investigation into Lingo-1's role in promoter and intragenic regions.

Overall, this article represents a well-conducted study that sheds light on the genetic and epigenetic factors associated with Lingo-1 in schizophrenia and their potential impact on cognitive function and white matter integrity. The article is suitable for publication pending minor revisions and addressing the suggestions below.

Minor Revisions:

  1. Clarify Abbreviations: Ensure that all abbreviations used in the article are defined upon their first use, including Lingo-1 and ASRB.

  2.  
  3. Improve Clarity: Some sentences and paragraphs could be rephrased for improved clarity. For example, consider simplifying complex sentence structures for easier comprehension.

  4.  
  5. Visual Presentation: While the figures are generally informative, ensure that all figure captions are complete and provide sufficient context for readers to understand the presented data.

  6.  
  7. Causality and Mechanistic Insights: The discussion could benefit from a more explicit exploration of the potential causality and mechanistic insights derived from the study findings. Provide hypotheses or mechanisms by which Lingo-1 genetic variations and methylation might affect cognitive deficits and white matter integrity.

  8.  
  9. Replication and Generalizability: Emphasize the need to replicate the findings in more extensive and diverse cohorts to enhance the generalizability and robustness of the results.

  10.  
  11. Ethical Considerations: Discuss ethical considerations related to genetic and epigenetic research, such as informed consent, data privacy, and the ethical implications of identifying potential markers for schizophrenia.

  12.  
  13. Conclusion: Summarize the key findings and their implications in a clear and concise conclusion section.

Overall, this study contributes valuable insights into the role of Lingo-1 in schizophrenia and cognitive function, and it is well-suited for publication with minor revisions and clarifications.

Author Response

Please see report attached. Thanks for your help

Round 2

Reviewer 2 Report

This manuscript reach to the quality of our journal.